# The Upper Airway Microbiota, Environmental Exposures, Inflammation, and Disease

**DOI:** 10.3390/medicina57080823

**Published:** 2021-08-14

**Authors:** Ziyad Elgamal, Pratyush Singh, Patrick Geraghty

**Affiliations:** 1Department of Biomedical Science, University of Guelph, Guelph, ON N1G 2W1, Canada; zelgamal@uoguelph.ca; 2Department of Medicine, Division of Pulmonary & Critical Care Medicine, State University of New York Downstate Medical Centre, Brooklyn, NY 11203, USA; 3Department of Biology, University of Western Ontario, London, ON N6A 5B7, Canada; psing64@uwo.ca

**Keywords:** upper airways, microbiota, dysbiosis, inflammation, airborne exposures, age, environmental exposure, disease

## Abstract

Along with playing vital roles in pathogen exclusion and immune system priming, the upper airways (UAs) and their microbiota are essential for myriad physiological functions such as conditioning and transferring inhaled air. Dysbiosis, a microbial imbalance, is linked with various diseases and significantly impedes the quality of one’s life. Daily inhaled exposures and/or underlying conditions contribute to adverse changes to the UA microbiota. Such variations in the microbial community exacerbate UA and pulmonary disorders via modulating inflammatory and immune pathways. Hence, exploring the UA microbiota’s role in maintaining homeostasis is imperative. The microbial composition and subsequent relationship with airborne exposures, inflammation, and disease are crucial for strategizing innovating UA diagnostics and therapeutics. The development of a healthy UA microbiota early in life contributes to normal respiratory development and function in the succeeding years. Although different UA cavities present a unique microbial profile, geriatrics have similar microbes across their UAs. This lost community segregation may contribute to inflammation and disease, as it stimulates disadvantageous microbial–microbial and microbial–host interactions. Varying inflammatory profiles are associated with specific microbial compositions, while the same is true for many disease conditions and environmental exposures. A shift in the microbial composition is also detected upon the administration of numerous therapeutics, highlighting other beneficial and adverse side effects. This review examines the role of the UA microbiota in achieving homeostasis, and the impact on the UAs of environmental airborne pollutants, inflammation, and disease.

## 1. Introduction

The upper airways (UAs) compose the respiratory system’s primary elements to conduct and process air to the lungs [1]. An average adult transfers over 7000 L of air daily through their UAs [1,2]. Hence, the airways have developed a complex immune system to filter the unsterile inhaled air [2,3]. Daily environmental contaminants, such as pollutants and pathogens, cause a shift in UA homeostasis, thereby activating the innate immune response and leading to a local inflammatory reaction [3,4]. The UAs harbor complex microbial communities to aid with their immense physiological responsibility. Similar to the gut microbiota, the upper airway symbionts are diverse, including pathogenic, commensal, and symbiotic microorganisms [2,5,6,7]. Collectively, the human body serves as a host for about 100 trillion microbes [7,8]. They endow us with crucial qualities such as aiding in metabolism, pathogen exclusion, and immune system training [5,7,9,10]. Dysbiosis, or loss of the beneficial and commensal microbes that prevent the colonization of opportunistic pathogens, is associated with local inflammation and exacerbation of many pulmonary disorders, such as chronic obstructive pulmonary disease (COPD), asthma, chronic rhinosinusitis (CRS), and various UA infections (UAIs). UAIs are a leading cause of loss of quality of life, and often death [11]. An average adult experiences two to four annual respiratory infections [11]. Consequently, UAIs constitute a yearly burden of 75 million physician visits, 150 million sick days away from work, and more than USD 10 billion in insurance costs, solely in American adults [11].

Several environmental factors and/or pre-existing conditions contribute to UA dysbiosis. This review examines the role of the UA microbiota in achieving homeostasis, and the impact of environmental airborne pollutants, inflammation, and disease on the UAs and their symbionts. It is also important to note that age, weight, and diet are also important factors for microbiota composition, diversity, and stability [12].

## 2. Upper Airway Microbial Niches

The UAs are defined as the nasal cavity, sinuses, nasopharynx, oropharynx, and laryngopharynx (Figure 1), all of which display variable niche parameters [6,13]. The development and maturation of the UAs is a complicated multistage process that occurs both pre- and postnatally [6,9,14,15]. The specialized resident microbial community plays a vital role in this post-natal development, as germ-free murine models indicate deranged UA physiology [6,9,14]. Microbial interactions lead to the development of highly specialized systems containing diverse niches across the UA cavities, with varying physiological functions [6,9,14,16]. For example, air-filled spaces such as the nasopharyngeal tonsils (adenoids), amongst others, serve as a major location for pathogen recognition and defense responses [17,18]. Each of these UA regions is subjected to specific microbial cellular and physiological gradients [14,19]. Differences in temperature, pH, mucus secretion, and relative oxygen concentration regulate bacterial colonization in the UAs [7,8,14,19].

The nasal cavity is constantly interacting with the external environment [18,20]. It is lined with diverse epithelial cell types, providing varying microenvironments for the microbiota, and leading to high microbial diversity in nasal niches [6,7,8,18,20]. Within the nasal cavity, the turbinates increase the surface area and thus regulate temperature and humidity [21]. As primary elements of the UAs, the anterior nares and nasal vestibule contain sebaceous glands and vibrissae to help moderate their dry and contaminated microhabitat [1,22]. The vibrissae filter large particles (>3 μm) from inhaled air, while smaller particles (0.5–3 μm) are trapped in the mucus layer [1,18,22].

Located adjacent to the nasal vestibule is the middle meatus [1,23]. This region serves as the drainage area for the UA sinuses (anterior ethmoids, maxillary, and frontal sinuses) [18,23]. The UA sinuses are air-filled cavities that are important for warming and humidifying inhaled air, as well as pathogen recognition and many other speculated functions [17,18,23]. These UA segments are composed of ciliated columnar epithelium, producing mucus that is then transported into the nasal cavity [18]. This mucosal drainage establishes local microhabitats in the nasal cavity with unique microbial populations [2,6,17,18]. Located at the ceiling of the nasal cavity is the olfactory area, which is composed of pseudostratified columnar olfactory epithelium that is modified to contain many neurons [6,21,24]. This area is also of interest, as there is a correlation between olfactory function and the composition of the local microbiota [24].

## 3. The UA Microbiota

Our body’s ecological microhabitats harbor all kinds of microorganisms, such as bacteria, viruses, and other eukaryotes [6,25]. Here, we will primarily restrict our discussion to bacteria in this review, since the bulk of UA microbiota research covers this kingdom. The development of a healthy UA microbiota depends on multiple innate factors such as genetics; however, lifestyle and environmental exposures also help determine the microbial composition [6,25]. Within healthy UAs, varying microbial species and biomass are observed across the nasal, nasopharyngeal, and oropharyngeal cavities [26]. The most abundant phyla in the human respiratory tract are *Proteobacteria*, *Firmicutes*, and *Bacteroidetes* [27]. However, the specific microbial composition may change across an individual’s lifetime (see Table 1) [8,22,25,28,29,30].

### 3.1. UA Microbiota in Infants

The microbiota develops within the first two years of life and subsequently ensures normal UA development and function [33]. Studies have shown that abnormal colonization of the UAs during infancy contributes to UA diseases such as asthma and other UAIs [33,34]. UA microbiota assemblage begins at birth, and resembles the environment exposed to during birth ((i.e., maternal vaginal (normal birth) or skin microbiota (cesarean)) [5,35]. If an infant is born via cesarean, their nasopharyngeal microbiota represents their mother’s skin microbiota, whereas if born via the vaginal route, their microbiota will resemble the maternal urogenital microbiota [5,35]. Following birth, the UA microbiota is maintained and nourished by feeding [5,36]. Breastfed infants support stable *Dolosigranulum*/*Corynebacterium* profiles, whereas formula-fed infants show increased *Staphylococcus aureus* profiles [5,36]. The microbiota of breast-fed infants seems to offer a stronger protective effect against UAIs [5,36]. Similarly, a *Moraxella* species (spp.)-dominated UA microbial profile in children has a protective effect against UAIs, yet there are exceptions, with some *Moraxella* spp. that facilitate viral-associated wheezing in young infants [9,33,34,35]. Additionally, increased nasopharyngeal signatures of *Streptococcus* can predict asthma in children [9,33,34,35]. Overall observations in the first 2 years of life suggest that the combination of *Dolosigranulum*, *Moraxella*, and *Corynebacterium* enhance protective respiratory physiology to a greater degree relative to a *Haemophilus* and *Streptococcus* dominated profiles in infants [35,37]. These latter profiles were associated with UAIs of viral and bacterial (*Haemophilus influenza* and *Streptococcus pneumoniae*) origin, as well as an elevated risk for development of respiratory-disease pathologies early in life [35,38,39].

### 3.2. UA Microbiota in Adults

In comparison to infants, a healthy adult’s UA microbiota has a higher bacterial load and lower diversity [9,19,40,41]. At the anterior nares, *Actinobacteria*, *Firmicutes*, and low levels of anaerobic *Bacteroidetes* have been detected (see Table 1) [5,24,40,42,43,44]. Different regions in the nasal cavity, such as the middle meatus, present a trumping diversity relative to the anterior nares [31]. The anterior nares are colonized by a greater proportion of *Firmicutes* and *Actinobacteria* relative to the middle meatus (Table 1) [31], whereas the middle meatus harbors more *Proteobacteria* in comparison to the anterior nares (Table 1) [31]. Contrastingly, the adult nasopharyngeal microbiota is similar to their anterior nares microbiota at a phylum level [19]. A healthy adult’s paranasal sinuses include bacterial genera such as *Staphylococcus*, *Corynebacterium*, and *Propionibacterium* (Table 1) [45]. Overall, the healthy adult UA microbiota is distinct, depending on the sampling site (middle meatus, oral cavity, oropharynx) [6,46].

### 3.3. UA Microbiota in the Geriatric Population

As humans age, the distinctive variations defining the microbiota in different regions of the UA gradually diminish [28]. These alterations in the microbial communities are thought to start between the ages of 40–65 years [9,28]. The nasal cavity microbiota that was once dominated by *Cutibacterium*, *Corynebacterium*, and *Staphylococcus* become (at >65 years) more representative of an oropharyngeal microbial community [9,28]. This age-associated melting pot in the UA microbiota may be explained as a consequence of the aging immune system [28], whereby the process of immunosenescence leads to increases in proinflammatory markers (i.e., TNF-α and IL-6) [47]. This may lead to loss of species richness, the opening of new environmental niches, and increased disease susceptibility [28,47]. As we age, the UA niches overlap, leading to increased microbial interactions (See Figure 2) [47]. Microbes continuously interact with the host, but also interact with other species living in their niches [48,49,50]. Such interactions are mostly beneficial; however, when niches overlap nutrients, there is increased competition, driving adverse interbacterial interactions [51,52].

## 4. UA Inflammation

Inflammation subjects the UA microbiota to selective pressure and may lead to dysbiosis, as it causes alterations in UA niches (see Figure 3) [58]. Clinical features of inflammation mostly stem from increased blood flow to the afflicted region in the UAs [58]. Such clinical features include increased vascular permeability, which facilitates the release of materials and immune cells, as well as increased mucus secretion and mucociliary clearance [58]. Even though the immune response plays a beneficial and protective role, altered inflammation may also lead to further harm, such as tissue injury and edema in the UAs, as well as altered niche parameters and dysbiosis [9,58].

The inflammatory immune response in the UAs is one of the multiple players, including cells such as granulocytes, lymphocytes, and antigen-presenting cells (APCs) [3,58]. Neutrophils are the most abundant granulocyte in the UAs, and their count is positively correlated with a higher degree of inflammation [3,58]. Neutrophilia affects resident *S. aureus* by causing the expression of more pathogenic profiles and the development of immune-evasion strategies [59,60]. Apart from direct inflammation, this also leads to an increase in mucus viscosity, changing the normal UA microhabitat and thereby contributing to UA dysbiosis [59,60,61]. Signatures of *Staphylococcus*, *Planococcaceae*, and *Lactococcus* were detected in the nasal microbiota of individuals with neutrophilia [62]. In addition, a higher neutrophil count in the mucosa is negatively correlated with microbial diversity [63]. Lymphocytes, which are also detected at higher levels during a period of inflammation, are associated with specific bacterial subsets [62]. A high lymphocyte count in nasal lavage samples was associated with increased colonization of *Staphylococcus*, *Rothia*, and *Enterococcus* within the nasal microbiota [62]. Conversely, the microbiota may also influence the host immune and inflammatory response. Commonly detected in the UA microbiota, *Propionibacterium* can produce bacteriocin, an antimicrobial peptide (AMP) that mitigates the inflammation associated with the overgrowth of pathogenic bacteria [64].

Noncellular mediators such as AMPs, cytokines, chemokines, immunoglobulins, arachidonic acid metabolites, and reactive oxygen species (ROS) also contribute to inflammation and thereby perturbate the UA microbiota. The nasal microbiota of subjects with increased concentrations of IL-6 is enriched with *Moraxella* [62]. High levels of IL-8 were associated with an increased presence of *Staphylococcus*, *Veillonella*, *Lachnospiraeae*, *Bacteroides*, and *Planococcaceae* in the nasal microbiota [62].

Disease states that drive UA chronic inflammation, such as cystic fibrosis (CF), obstructive sleep apnea, CRS, and asthma, also impact the UA inflammatory profile and microbiota [65]. CF patients with higher colonization of *S. aureus* within their UAs had distinct levels of inflammatory biomarkers in nasal lavage samples [65]. Notable levels of IL-6, IL-8, IL-1β, MMP-9, and neutrophil elastase were detected [65]. This suggests that increased colonization of *S. aureus* in diseased UAs could further contribute to the inflammatory response [65]. *S. aureus* in low-infection colony numbers triggers the secretion of the anti-inflammatory cytokine IL-10, yet when present in higher colony numbers, it reduces IL-10, leading to a more proinflammatory response [65,66,67].

## 5. Environmental Exposures and the UA Microbiota

Acute changes in environmental factors cause perturbations to the UA microbiota [30,68,69]. In this section, we will define the main environmental exposures that contribute to UA microbiota dysbiosis.

### 5.1. Seasonal Influences

Seasons contribute to changes in the UA microbiota [69]. Infants and children presented a higher incidence of UAI and *S. pneumoniae* colonization throughout winter [70,71,72]. Wintertime also increased the abundance of *Proteobacteria* and increased carriage of *Fusobacteria* and *Cyanobacteria* [73], whereas infants with respiratory infections have increased carriage of *Haemophilus* across spring and summer [33]. During the autumn–winter seasons, infants with respiratory infections presented with decreased carriage of *Moraxella* [33]. In adults, increased presence of *M. catarrhalis* and coronavirus is detected during the winter, and *Klebsiella pneumoniae* presents with higher densities in the summertime [72].

### 5.2. Air Pollution

Air pollution, which is a mixture of harmful gases and particulate matter (PM), is a major contributor to dysbiosis and many respiratory diseases [74,75]. Globally, almost 90% of people live in areas with suboptimal air quality, according to the WHO. Air pollutants, such as nitrogen and sulfur oxides, as well as ozone, directly irritate the UA epithelium by causing increased oxidative stress and local inflammation [76]. Carcinogens such as diesel emissions and household pollutants arise from burning fuels such as wood and coal [77,78]. PM are foreign substances (i.e., a mixture of solid particles and liquid droplets) found in air that are defined based on size and aerodynamic qualities [79,80]. Larger PM (~10 μm) may get stuck in the UA, while smaller PM (~4 μm) is more likely to travel down the bronchioles and alveoli [79,80]. Short-term exposure to PM influenced the nasal microbiota of 40 healthy subjects by decreasing indices of bacterial diversity, which resulted in increased susceptibility to UAIs (Table 2) [81]. Similarly, exposure to high concentrations of PM for three consecutive days was followed by an increased pharyngeal abundance of particular taxa (Table 2) [82]. Air pollution (PM_2.5_ and NO_2_) is associated with Ružička dissimilarity and the abundance of *Corynebacteriaceae* (NO_2_ only) in healthy infants [83].

### 5.3. Cigarette Smoke Exposure

Cigarette smoke is a major contributor to disease and directly contacts the UA surfaces [30,82,86,90,91]. It directly impacts the microbiota; for example, causing oxidation of antimicrobial activity [82]. This change in niche parameters drives the colonization of Gram-positive anaerobic microbes (Table 2) in a smoker’s nasopharynx [30]. The recolonization of smokers’ UAs may also include pathogens associated with UAIs and endocarditis [30]. Notably, a cigarette may also serve as a source of opportunistic pathogens, as some commercially available cigarettes have bacterial signatures (Table 2) [86]. Passive smokers also suffered from smoke-induced dysbiosis (Table 2) [84,85]. Higher levels of *S. pneumoniae* were detected in children with smoking parents [86]. There was also an observed increase in the prevalence of various UAIs [82,87,88]. Cigarette smoke enhances bacterial attachment, colonization, and biofilm formation in various mechanisms, including the increased production of the bacterial fimbrial protein, FimA [29,82,92,93]. Cigarette smoke alters host physiological functions such as mucociliary clearance, which in turn impairs the innate immune response to pathogens and the UA microenvironments [32,91,94]. Fortunately, smoke-induced dysbiosis can be alleviated by smoke cessation for approximately one year [29,82,85]. Partial recovery of UA microbiota and physiology was also detected [30,82,95].

## 6. Disease Pathology and Dysbiosis

### 6.1. UA Diseases and Dysbiosis

CRS, defined as inflammation within the UA sinuses for more than a period of 12 weeks, is well studied in relation to the UA microbiota [96]. In CRS patients, there was a decrease in UA microbial diversity, and their nasal microbiota was frequently dominated by coagulase-negative bacteria (Table 3) [45,97,98]. Increased prevalence of anaerobic pockets in UA niches of CRS patients may explain the decline in microbial diversity and increased presence of anaerobic bacteria [98,99,100]. *Streptococcic*-dominated UAs evoked proinflammatory, T-helper 1 (TH1) responses and encoded an ansamycin biosynthesis gene pathway [101]. Similarly, increased colonization of *Pseudomonadaceae* evoked a TH1 proinflammatory response and activated tryptophan metabolism gene pathways instead [101]. The development of nasal polyps was correlated with increased colonization of Corynebacteriaceae and enhanced IL-15 expression [101].

Additionally, the nasal microbiota potentially plays a vital role in regulating the local immune response, and consequently the pathophysiology of allergic rhinitis (AR) [99]. Samples from the middle meatus and nasal vestibule showed varying microbial profiles in AR patients vs. healthy controls [117]. Spring pollen leads to eosinophilia and increased AR symptoms, and is associated with higher middle meatus microbial biodiversity [99].

Genetic disorders such as CF and primary ciliary dyskinesia (CD) also impact the UA microbiota, as they affect an individual’s mucociliary clearance [118,119]. Children with CF present with thickening of the UA mucus layer and increased *Staphylococcus* spp., while other beneficial bacteria in their microbiota decrease in abundance [37,119]. CD is a combination of disorders that result in deranged ciliary action and decreased ability to clear mucus from the UA. People presenting with CD frequently suffer from CRS and are at increased risk of UAI and asthma development [118]. Equally, CF patients frequently develop CRS with the presence of nasal polyps [119].

Recently studies have proved that the nasal microbiota has a role in the onset and severity of asthma [33]. Distinct nasopharyngeal microbiota predicts the risk and severity of asthma-related inflammation [33]. During the first year of life, increased nasopharyngeal colonization of *Streptococcus* spp. is a strong predictor of future asthma development [33], and the nasopharyngeal microbiota in children with asthma differs from healthy controls (Table 3) [120]. Adults with exacerbated and non-exacerbated asthma have distinguished nasal microbiota [104,105]. Taxa from *Bacteroidetes* and *Proteobacteria* dominated the nasal microbiota of asthma patients (Table 3) [117,121]. Bacterial species abundance and glycerolipid metabolism varied depending on asthma state (Table 3) [103,104]. Similarly, individuals suffering from obstructive sleep apnea (OSA) have a distinct nasal microbiota [62]. Neutrophilic samples were enriched with *Planococcaceae*, *Lactococcus*, and *Staphylococcus,* while samples that had a high lymphocyte count or higher levels of IL-8 were enriched with *Streptococcus*, *Rothia*, and *Enterococcus*. Such alterations were also not impacted by three months of treatment of continuous positive airway pressure. Hence, the microbial diversity and composition distinctions in OSA patients correlated with inflammatory biomarkers.

### 6.2. UAIs and Dysbiosis

Different UAIs correlate with specific microbial profiles (Table 3). Positive and negative changes to the microbiota may be attributed to intramicrobial interaction and the immune response against a pathogen [122,123,124]. Influenza A virus infections modify the UA microbiota by increasing the presence of pathogenic bacteria [123,124]. Influenza virus infections in adults are associated with increased nasal carriage of *Streptococcus pneumoniae* and *S. aureus* [106]. *S. pneumoniae* secrete proteases that stimulate viral hemagglutinin activation and may also modulate the host immune response to facilitate influenza A viral infection [125,126]. Elevated *S. pneumoniae* density in the UAs increases the risk of pneumococcal pneumoniae and is associated with influenza virus, rhinovirus, and adenovirus infections [109]. In infants, a nasopharyngeal microbiota dominated by *Haemophilus* presented higher *rhinovirus-A* spp. infections [110]. *Rhinovirus-C* spp. infections are more likely to occur in infants with a *Moraxella*-dominated nasopharynx [110]. Moreover, a predominant *Haemophilus* nasopharyngeal microbiota is associated with delayed clearance of respiratory syncytial virus in infants hospitalized for bronchitis [127]. Equally, in health infants with human rhinovirus (HRV) infections, a study found a significant difference in the Shannon diversity index between rare and frequently infected HRV groups, and higher bacteria density in the frequently infected group [128].

## 7. UA Microbiota Therapeutics

Unsurprisingly, multiple therapeutics impact the UA microbiota, such as saline rinses, intranasal corticosteroids, probiotics, and antibiotics. Treatments such as antibiotics directly impact the microbiota with their antimicrobial properties and lead to a decline in UA microbial diversity and biomass [129,130]. The administration of antibiotics such as beta-lactams and mupirocin are associated with increases in Gram-negative bacteria [33,129,131,132]. Additionally, some commensal bacteria can tolerate such conditions and become pathogenic following antibiotic use [133,134]. This shift in the UA microbiota due to antibiotic treatment is assumed to last for a minimum of two weeks post-treatment, according to samples from the anterior nares [134]. Intranasal corticosteroids such as mometasone furoate monohydrate also decrease the biomass and biodiversity of the UA microbiota [129,130,135]. Intranasal administration of corticosteroids correlates with decreases in the overall UA microbial biomass by suppressing several taxa (i.e., *Moraxella* spp., *streptococci*) whilst promoting others, such as *Staphylococci* [129,130,135]. Intranasal vaccinations also play a role in disturbing the UA microbiota. Specifically, vaccination against *S. pneumoniae* alters the carriage of *S. aureus* and *H. influenzae*, yet *S. pneumoniae* colonization remains unchanged [136]. Intranasal live attenuated influenza vaccine significantly increased the nasal microbial taxa richness and increased the variation in influenza-specific IgA production [137]. Nasal rinses are used today to prevent and treat UAIs, and are typically a saline mixture with variable pH, but could also be as simple as distilled or tap water [138]. The inexpensive, simple method has a very low direct impact on the UA microbiota [138]. However, in some cases, contamination due to the use of tap or well water has been shown to lead to mycobacterial infections [138,139,140,141]. Surgical procedures in the UAs used to treat refractory sinusitis and polyposis also impact the microbiota [138]. The procedure enlarges the sinus ostia, improves mucociliary clearance, and facilitates topical therapy access [142]. It influences UA physiology by reducing the ambient temperature and humidity within the airways [142,143]. Patients are prescribed antibiotic and/or probiotic adjuvant therapeutics to prevent pathologic recolonization within the UAs, and the procedural outcomes are mostly positive [131,143].

Probiotics are a promising therapeutic to dysbiosis and are used in many diseases, such as asthma and CRS [144]. Probiotics are living beneficial bacteria that are administered directly into the nasal cavity to provide health benefits to the host [141,144,145]. However, the efficacy of probiotics is dependent on their ability to colonize specific regions of the UA epithelium [146]. Recently, probiotics were suggested as an adjuvant therapy to antimicrobial treatments to help mitigate their negative effects on microbial diversity and biomass [144,146]. They are sometimes referred to as keystone species due to their ability to act as pioneers and help re-establish the normal UA microbiota [147]. In turn, probiotics induce adaptations that improve the epithelial barrier and its function [144,146,148,149]. Positive interactions with host immune components induce physiological improvement of the UA epithelium by regulating signaling transduction pathways [144,146,148,149]. The residing UA symbionts interact with probiotic species, increasing antimicrobial production and inhibiting pathogen overgrowth via changing pH of the niche [147,150,151]. Notably, *Enterococcus faecalis* administration in children with acute sinusitis reduces the frequency and duration of disease [146,152]. Probiotics possess immunomodulatory functions, including regulation of circulating lymphocytes and cytokines [153,154,155,156]. *Lactobacillus rhamnosus* exemplifies this by increasing Th1 cells and decreasing Th2 cells in rodent models [157,158]. Moreover, probiotics modulate inflammatory reactions against specific immunogens by regulating lymphocyte responses and mucosal IgA levels [159].

## 8. Limitations and Future Research

Most of the UA microbiota knowledge is obtained through cultivation assays and next-generation sequencing. Both short reads provide basic information about the UA microbial diversity and taxonomic composition of the UA microbiota. Overcoming said limitations requires long-read technologies such as Oxford Nanopore [160] or Pacific Bioscience technology [161]. These will permit the sequencing of the entire 16S rRNA gene and provide more accurate results in terms of species- or strain-level profiling. Shotgun metagenomics offers insight into the microbiota and its functions. This technology also allows the assembly of draft uncultured microbial genomes associated with human health or disease. Sampling of the disease-afflicted region improves study findings. Positively correlated microbial candidates with both the site and incidence of a disease may help relieve its burden if removed. In opposition, negatively correlated disease candidates may play a protective role and may be administered as probiotics. Hence, it is crucial to address the limitations of sampling from less-accessible cavities of the UAs; appropriate sampling tools will minimize contamination from neighboring sites. Intramicrobial and microbial-host interactions are also research areas that require more attention. Since they will benefit our research designs in the pursuit of therapeutics and causal connection with UA disorders. Nonetheless, proving causality and displaying the efficacy and feasibility of treatments requires approaches such as animal model development and arranging more clinical trials. Finally, we have focused mostly on bacterial changes in the UA microbiota, but emerging evidence has revealed the composition of the lung virome, the global viral communities present in the airways [162]. Equally, recent data suggests that the microbiota compositions in critically severe COVID-19 patients were likely due to intubation and mechanical ventilation [111]. A recent study in children presenting with a positive SARS-CoV-2 infection demonstrated that both the upper respiratory tract and the gut microbiota were altered. The alteration of the microbiota in these children was dominated by the genus *Pseudomonas* (see Table 3), and remained altered up to 25–58 days in different individuals [112]. As children do not experience the complications associated with adult COVID-19, these microbiota profiles may give important insight into the role of the microbiota in disease susceptibility. This same group also demonstrated that bacterial diversity was lower in adult COVID-19 patients than healthy controls [113]. Importantly, microbiota diversity was greater enriched in mild COVID-19 patients, which suggests that a more complex microbiota may aid in recovery from COVID-19. A recent study in the USA also observed significant microbiota changes in an adult COVID-19 population [114]. Several UA immune responses are linked to COVID-19 outcomes, including changes in human epididymis secretory protein 4 [163]. However, further analysis of COVID-19′s influence on the UA microbiota and the influence of the UA microbiota on COVID-19 severity need further addressing. For further reading on microbiota and immune responses in COVID-19 upper airway, we recommend the review by Di Stadio *et al*. [164].

## 9. Conclusions

Overall, the UA microbiota literature suggests that the development of a diverse microbiota across the UAs aids in maintaining healthy and disease-free UAs. Age, diet, environmental exposures, infections, and therapeutics all contribute to UA microbiota and health. The loss of UA community segregation appears to contribute to inflammation and disease that results in disadvantageous microbial–microbial and microbial–host interactions. Maintaining UA microbiota homeostasis may represent a major approach for future therapeutics. New approaches to sampling and analysing microbiota diversity will aid in our understanding of UA diseases. Future scientific advancements to recover the healthy microbiota may become a common choice in the treatment of patients with UA diseases.

## Figures and Tables

**Figure 1 medicina-57-00823-f001:**
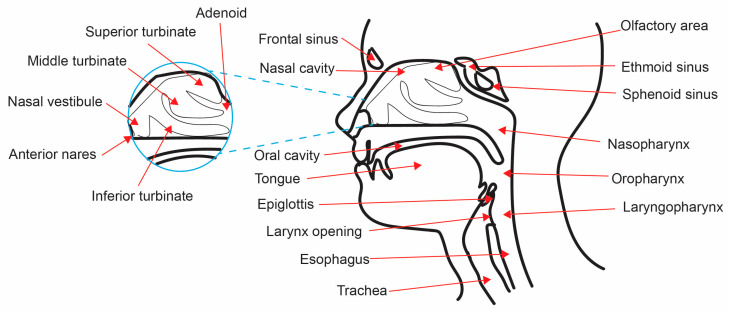
The anatomy of the upper airways. The UAs are formed by many different cavities, including the nasal cavity, sinuses, nasopharynx, oropharynx, and laryngopharynx.

**Figure 2 medicina-57-00823-f002:**
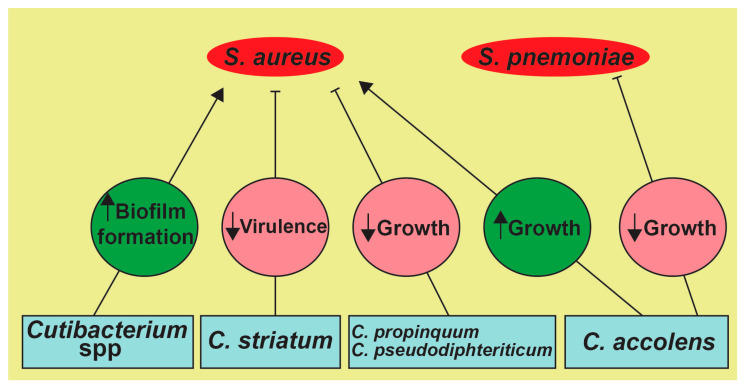
Possible microbial interactions in the UAs. The secretions of *Staphylococcus lugdunesis* (i.e., lugdunin, a thiazolidine) inhibits the growth of *S. aureus* [53]. Similarly, *Corynebacterium* spp. shifts *S. aureus* towards a more commensal state by attenuating its virulence components, such as the agr operon and genes involved in its hemolytic activity [50,54,55]. Moreover, *Corynebacterium* and *Cutibacterium* spp. affect the growth of *S. aureus*, as the most commonly secreted porphyrin by *Cutibacterium* spp. facilitates *S. aureus* aggregation and biofilm formation with other UA microbes [52,56,57].

**Figure 3 medicina-57-00823-f003:**
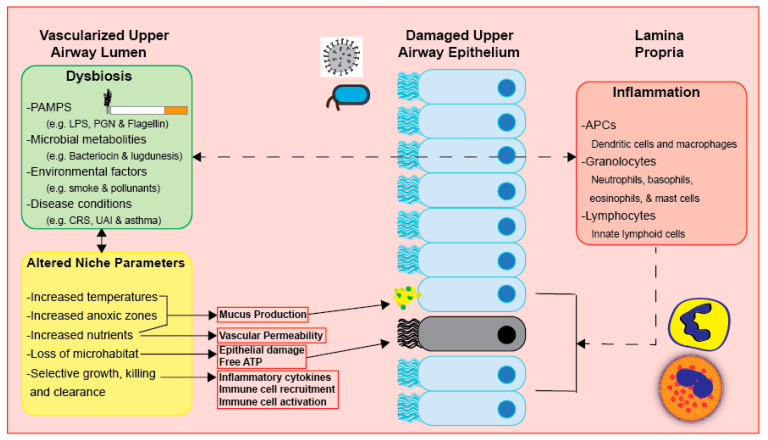
Dysbiosis, inflammation, and altered niche parameters heavily impact respiratory physiology. These three factors are at continuous interplay. However, the direct mechanisms that may further elucidate this three-way relationship have not been defined.

**Table 1 medicina-57-00823-t001:** Detected UA bacteria across age in a healthy population.

Healthy Population	Study	Sampling Site and Collection Method	Detected Bacteria
Infants	[26]	Anterior nares Nasal swab	*Corynebacterium, Propionibacterium, Bifidobacterium, Streptococcus, Staphylococcus, Dolosigranulum, Moraxella*
	[26]	Nasopharynx Nasopharyngeal swab	*Corynebacterium, Propionibacterium, Bifidobacterium, Bacteroides, Staphylococcus, Faecalibacterium, Streptococcus, Moraxella*
	[26]	Oropharynx Oropharyngeal swab	*Prevotella, Streptococcus*, *Vaillonella, Haemophilus*, *Moraxella*, *Neisseria*
Adults	[31]	Anterior nares Nasal swab	*Corynebacterium, Propionibacterium, Prevotella, Dolosigranulum*, *Staphylococcus*, *Streptococcus*, *Moraxella*, *Escherichia shigella*
	[31]	Middle meatus Middle meatus swab	*Corynebacterium, Propionibacterium, Prevotella*, *Dolosigranulum*, *Staphylococcus*, *Streptococcus*, *Moraxella*, *Escherichia shigella*
	[32]	Sinus Sinus swab	*Corynebacterium, Propionibacterium*, *Prevotella, Staphylococcus*, *Anaerococcus*, *Peptoniphilus*, *Ralstonia*
	[9]	Nasopharynx Nasopharyngeal swab	*Corynebacterium, Propionibacterium*, *Bifidobacterium*, *Prevotella*, *Sphingobacterium*, *Staphylococcus*, *Faecalibacterium*, *Streptococcus, Pseudomonas*, *Haemophilus*
	[9]	Oropharynx Oropharyngeal swab	*Corynebacterium, Rothia, Prevotella*, *Porphyromonas, Streptococcus*, *Vaillonella*, *Haemophilus*, *Moraxella*
Elderly	[28]	Anterior nares Nasal swab	*Corynebacterium, Propionibacterium, Bifidobacterium, Prevotella*, *Bacteroides, Streptococcus*, *Staphylococcus*, *Vaillonella, Moraxella*, *Pseudomonas*
	[28]	Nasopharynx Nasopharyngeal swab	*Corynebacterium, Propionibacterium, Bifidobacterium, Prevotella*, *Bacteroides, Streptococcus*, *Staphylococcus*, *Vaillonella*, *Moraxella*, *Pseudomonas*
	[28]	Oropharynx Oropharyngeal swab	*Corynebacterium, Propionibacterium, Bifidobacterium, Prevotella*, *Bacteroides, Streptococcus*, *Staphylococcus*, *Vaillonella, Moraxella*, *Pseudomonas*

**Table 2 medicina-57-00823-t002:** Environmental exposures and the changes in the UA microbiota.

Exposure	Study	Population and Sampling Site	Detected Bacteria
Active cigarette smoke	[30]	Adults Nasopharynx	↓ *Actinomycetaceae*, ↓ *Corynebacteriaceae*, ↓ *Coriobacteriaceae*, ↑ *Eggerthella*, ↓ *Flexibacteriaceae*, ↓ *Flavobacteriaceae*, ↑ *Porphromonadaceae*, ↓ *Leuconostocaceae*, ↑ *Erysipelotrichaceae*, ↑ *Aerococcaceae*, ↑ *Eubacteriaceae*, ↑ *Incertae Sedis XIII*, ↑ *Peptostretococcaceae*, ↑ *Ruminococcaceae*, ↑ *Lachnospiraceae I.S.* spp., ↑ *Anaerovorax*, ↑ *Dorea,* ↑ *Erysipelotrichaceae I.S*., ↑ *Eubacterium* spp., ↑ *Abiotrophia* spp., ↓ *Rhodocyclaceae*, ↓ *Rhodobacteraceae*, ↓ *Enterobacteriaceae*, ↓ *Alcalgenaceae*, ↓ *Methylophilacea*, ↓ *Shigella* spp., ↑ *Pasteurellaceae*, ↑ *Haemophilus* spp.
	[84]	Adults Nasopharynx	↑ *Streptococcus pneumoniae*, ↑ *Streptococcus pyogenes,* ↑ *H. influenzae*, ↑ *M. catarrhalis*
Passive cigarette smoke	[85]	Infants (<60 months) in the USA Nasopharynx	↑ *Streptococcus pneumoniae*
Tobacco samples from cigarettes	[86]	Cigarettes	*Bacillus*, *Clostridium*, *Enterococcus*, *Staphylococcus, Acinetobacter*, *Burkholderia*, *Klebsiella*, *Pseudomonas aeruginosa*, *Serratia*, *Campylobacter*, *Proteus*
Air pollutants: PM_2.5_, PM_10_, SO_2_, NO, O_3_	[87]	Healthy young adults in northeastern China Oropharynx	↓ *Prevotella*, ↓ *Fusobacterium*, ↓ *Capnocytophaga,* ↓ *Veillonella,* ↓ *Campylobacter*
Air pollutants: PM2.5, NO_2_	[83]	Healthy infants Nasal swabs	↓ *Corynebacteriaceae* (NO_2_ only)
Air pollutants: PM_2.5_, PM_10_	[81]	Healthy subjects in Italy Nasal cavity	(−) *Rothia*, (−) *Corynebacterium,* (−) *Flavobacterium,* (−) *Finegoldia*, (−) *Oribacterium*, (−) *Streptococcus,* (−) *Neisseria*, (−) *Haemophilus*, (−) *Actinobacillus*, (+) *Moraxella*
	[82]	Vendors at an open-air farmer’s market in China Oropharynx	↑ *Leptotrichia*, ↓ *Prevotella 7,* ↑ *Streptococcus*, ↑ *Staphylococcus,* ↑ *Haemophilus*, ↑ *Moraxella*
Air pollutants: O_3_	[88]	Healthy young adults in China Nasal cavity	↑ *Moraxellaceae*, ↑ *Pseudomonadaceae*
Household air pollutants	[89]	Healthy adults in Malawi Nasal cavity	↓ *Tropheryma,* ↑ *Streptococcus,* ↓ *Neisseria*, ↑ *Petrobacter*

Symbols: (↑ or ↓) represent the higher or lower presence of said bacteria detected upon exposure, respectively, while (− and +) indicate a negative or positive correlation with the said disease condition, respectively.

**Table 3 medicina-57-00823-t003:** Disease status determines the UA microbial composition.

Exposure	Study	Population and Sampling Site	Detected Bacteria
Chronic Rhinosinusitis (CRS)	[99]	Adults Middle meatus	↑ *Corynebacterium*, ↑ *Curtobacteria*, ↑ *Staphylococcus*, ↑ *Pseudomonas*
	[102]	Adults Sinus	*Corynebacteriaceae, Staphylococcaceae*, *Streptococcaceae, Pseudomonadaceae*
	[100]	Adults with nasal polyps Middle meatus	(+) *Fusobacterium*, (+) *Streptococcus*, (+) *Haemophilus*
	[100]	Adults without nasal polyps Middle meatus	*Corynebacterium, Staphylococcus*, *Alloiococcus*
	[101]	Adults Middle Meatus	↓ *Corynebacterium,* ↑ *Porphyromonas*, ↑ *Prevotella,* ↑ *Anaerococcus*, ↑ *Lactobacillus*, ↑ *Finegoldia*, ↑ *Peptoniphilus*, ↑ *Dialister*, ↑ *Parvimonas*, ↓ *Staphylococcus*, ↓ *Dolosigranulum*
	[103]	Adults Nasopharynx	↓ *Corynebacterium*, ↓ *Propionibacterium,* ↓ *Anaerococcus*, ↓ *Finegoldia*, ↓ *Peptoniphilus*, ↓ *Staphylococcus*
Asthma	[104]	Adults (18–45 years old) in Korea Nasopharynx	↓ *Corynebacterium,* ↓ *Moraxella*
Non-Exacerbated Asthma	[105]	Adults (mean age 36 years old) Nasopharynx	↑ *Dialister*
Exacerbated Asthma	[105]	Adults (mean age 36 years old) Nasopharynx	↑ *Gardnerella,* ↑ *Prevotella,* ↑ *Alkanindiges*
Obstructive Sleep Apnea	[62]	Adults in the USA and Spain Nasopharynx	↑ *Streptococcus,* ↑ *Prevotella,* ↑ *Veillonella*
Influenza A Virus	[106]	Infants (under the age of 2 years old) Oropharynx	↑ *Streptococcus pneumoniae*, ↑ *Staphylococcus aureus*
	[107]	Patients (up to 52 years old) Nasopharynx	↑ *Corynebacterium* spp., ↑ *Streptococcus,* ↓ *Pseudomonas*
	[108]	Children in China Nasopharynx/Oropharynx	↑ *Dolosigranulum* spp., ↑ *Staphylococcus aureus,* ↑ *Phyllobacterium* spp., *↑ Moraxella* spp.
Adenovirus	[109]	Patients (mean age 24–25 years) in South Africa Nasopharynx	↑ *Streptococcus pneumoniae*
Respiratory Syncytial Virus	[110]	Hospitalized infants (under 1 year of age) in the USA Nasopharynx	↑ *Haemophilus* spp.
SARS-CoV-2	[111]	Elderly patients in China Oropharynx	↑ *Klebsiella pneumoniae,* ↑ *Enterococcus* spp., ↑ *Staphylococcus*
	[112]	Children (7–139 months old) Posterior nasopharynx and oropharynx	↓ *Bacteroidetes*, ↓ *Firmicutes*, ↑ *Proteobacteria*, ↑ *Pseudomonas*, ↑ *Herbaspirillum*, ↑ *Burkholderia*, and ↑ *Comamonadaceae_U*
	[113]	Adults (17–68 years old) Throat swabs	↑ *Bacteroidales*, ↑ *Fusobacterium*, ↑ *Porphyromonas*, ↑ *Prevotella*, and ↑ *Neisseria*
	[114]	>18-year-old, nonhospitalized COVID-19 patients Midturbinate swab	↑ *Peptoniphilus lacrimalis*, ↑ *Campylobacter hominis*, ↑ *Prevotella 9 copri*, ↑ *Anaerococcus*, ↓ *Corynebacterium*, ↓ *Staphylococcus haemolyticus*, ↓ *Prevotella disiens*, and ↓ *Corynebacterium*
Middle East Respiratory Syndrome (MERS) Coronavirus	[115]	Adults (47–74 years old) in Saudi Arabia Oropharyngeal swabs and tracheal aspirates	(+) *Acinetobacter baumannii*, (+) *Pseudomonas aeruginosa*, and (+) *Streptococcus pneumoniae*
	[116]	Young children (under 3 years old) in Peru Nasopharynx	↑ *Streptococcus pneumoniae*, ↑ *Staphylococcus aureus*
Rhinovirus-A	[109]	Patients (mean age 24–25 years) in South Africa Nasopharynx	↑ *Streptococcus pneumoniae*, ↑ *Staphylococcus aureus*
	[110]	Hospitalized infants (under 1 year of age) in the USA Nasopharynx	↑ *Haemophilus* spp.
Rhinovirus-C	[116]	Young children (under 3 years old) in Peru Nasopharynx	↑ *Streptococcus pneumoniae*, ↑ *Staphylococcus aureus*
	[109]	Patients (mean age 24–25 years) in South Africa Nasopharynx	↑ *Streptococcus pneumoniae*, ↑ *Staphylococcus aureus*
	[110]	Hospitalized infants (under 1 year of age) in the USA Nasopharynx	*↑ Moraxella* spp.

Symbols: (↑ or ↓) represent the higher or lower presence of said bacteria detected upon exposure, respectively, while (+) indicate a positive correlation with the said disease condition, respectively.

## Data Availability

Not applicable.

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
