# Peer review of "The Upper Airway Microbiota, Environmental Exposures, Inflammation, and Disease"

_medicina, 2021, doi:10.3390/medicina57080823_

Round 1
Reviewer 1 Report
The authors were aimed to examine the role of the upper airways (UA) microbiome in achieving homeostasis, and the impact of environmental airborne pollutants, inflammation, and disease on the UA and its symbionts.
The study covers some issues that have been overlooked in other similar topics. The structure of the manuscript appears adequate and well divided in the sub-paragraphs. Moreover, the study is easy to follow, but some issues should be improved before publication.
The manuscript needs moderate English change and grammar correction. Please also check typos thorough the text.
Introduction section: the general characteristics of the normal microbiota in the different anatomical sites of the airways have been reported in relation to some factors such as the effect of age, diet and others on its composition and stability. Will be useful to the readers o stress better those concepts (please see and discuss: DOI: 10.3390/biology9100318; doi: 10.3390/biology9120415).
Conclusion paragraph required a general revision to eliminate redundant sentences and to add some "take-home message".
Author Response
The manuscript needs moderate English change and grammar correction. Please also check typos thorough the text.
Response: We have corrected the grammatical errors and typos throughout the manuscript
Introduction section: the general characteristics of the normal microbiota in the different anatomical sites of the airways have been reported in relation to some factors such as the effect of age, diet and others on its composition and stability. Will be useful to the readers o stress better those concepts (please see and discuss: DOI: 10.3390/biology9100318; doi: 10.3390/biology9120415).
Response: We have now added the following sentence to the introduction, “It is also important to note that age, weight, and diet are also important factors for microbiota composition, diversity, and stability”. We have included the recommended review paper reference as further reading for the readers for metabolism and gut microbiota.
Conclusion paragraph required a general revision to eliminate redundant sentences and to add some "take-home message".
Response: We have expanded on the conclusion paragraph as recommended by the reviewer. See lines 409-417
Reviewer 2 Report
The submitted manuscript is aimed to summarize the role of the respiratory microbiome and some environmental factors on humans.
The text is well organized and sounding, but lacks some important informations, for example the role of respiratory microbiota and environment in COVID-19 pathophysiology.
Major concerns:
Although the terms microbiome and microbiota are often used as synonims, in the context of this manuscript the term microbiota is preferable. Please change it thorough the text.
In addition, to make the ms more interesting for a wider readership, the authors:
- should discuss more about SARS-CoV-2 and respiratory microbiota
- should add more information from clinical trials about the reciprocal influence of microbiota, pollutions and health status. Please read and discuss the following PMIDs: 34256075, 30870661, 29518345, 33019595, 33061464, 33147871, 33605840, 34287009, 31276660, 33504318, 33596252, 34313463, 31983528.
Minor concerns:
several typos in the text must be amended (i.e., shotgun metagenomics, not shut-gun metagenomics; Streptococcus pneumoniae, not Streptococcus pneumonia)
authors must use italics for bacterial names, both in the text and tables
Author Response
The text is well organized and sounding, but lacks some important informations, for example the role of respiratory microbiota and environment in COVID-19 pathophysiology.
Response: We have added the following text to the manuscript, which can be found on lines 388-407:
“Finally, we have focused mostly on bacterial changes in the UA microbiota but emerging evidence has revealed the composition of the lung virome, the global viral communities present in the airways [155]. Equally, recent data suggests that the microbiota compositions in critically severe COVID-19 patients were likely due to intubation and mechanical ventilation [156]. A recent study in children presenting with a positive SARS-CoV-2 infection demonstrated that both the upper respiratory tract and the gut microbiota were altered. The alteration of the microbiota in these children was dominated by the genus Pseudomonas (see Table 3), and remained altered up to 25–58 days in different individuals [157]. As children do not experience the complications associated with adult COVID-19, these microbiota profiles may give important insight into the role of the microbiota in disease susceptibility. This same group also demonstrate that bacterial diversity was lower in adult COVID-19 patients than healthy controls [158]. Importantly, microbiota diversity was greater enriched in mild COVID-19 patients that suggests that a more complex microbiota may aid in recovery from COVID-19. A recent study in the USA also observed significant microbiota changes in an adult COVID-19 population [159]. Several UA immune responses are linked to COVID-19 outcomes, including changes in human epididymis secretory protein 4 [160]. However, further analysis on COVID-19 influence on the UA microbiota and the influence of the UA microbiota on COVID-19 severity need further addressing. For further reading on microbiota and immune responses in COVID-19 upper airway, we recommend the review by Di Stadio et al. [161].” and in Table 3.
Major concerns:
Although the terms microbiome and microbiota are often used as synonims, in the context of this manuscript the term microbiota is preferable. Please change it thorough the text.
Response: We have made this change throughout the manuscript.
In addition, to make the ms more interesting for a wider readership, the authors:
should discuss more about SARS-CoV-2 and respiratory microbiota
should add more information from clinical trials about the reciprocal influence of microbiota, pollutions and health status. Please read and discuss the following PMIDs: 34256075, 30870661, 29518345, 33019595, 33061464, 33147871, 33605840, 34287009, 31276660, 33504318, 33596252, 34313463, 31983528.
Response: We have expanded on the COVID-19 section, which can be found on lines 388-407. We thank the reviewer for their thorough literature search and article recommendations. We have included several of these recommended references and discussed them on lines 236-238, 317-320, 388-407, and in tables 2 and 3. Several of the references were specific to lower airways or were review papers and we did not add them here due to being outside of the topic parameters.
Minor concerns:
several typos in the text must be amended (i.e., shotgun metagenomics, not shut-gun metagenomics; Streptococcus pneumoniae, not Streptococcus pneumonia)
authors must use italics for bacterial names, both in the text and tables
Response: We have corrected the grammar mistakes and typos